# Immigration as a Divisive Topic: Clusters and Content Diffusion in the Italian Twitter Debate

**Salvatore Vilella [1,2,\*], Mirko Lai [1], Daniela Paolotti [2] and Giancarlo Ruffo [1]**

[1]   Department of Computer Science, University of Turin, 10149 Torino, Italy; mirko.lai@unito.it (M.L.);
      giancarlo.ruffo@unito.it (G.R.)
[2]   ISI Foundation, 10126 Torino, Italy; daniela.paolotti@isi.it
[\*]   Correspondence: salvatore.vilella@unito.it

**Abstract:** In this work, we apply network science to analyse almost 6 M tweets about the debate around immigration in Italy, collected between 2018 and 2019, when many related events captured media outlets' attention. Our aim was to better understand the dynamics underlying the interactions on social media on such a delicate and divisive topic, which are the actors that are leading the discussion, and whose messages have the highest chance to reach out the majority of the accounts that are following the debate. The debate on Twitter is represented with networks; we provide a characterisation of the main clusters by looking at the highest in-degree nodes in each one and by analysing the text of the tweets of all the users. We find a strongly segregated network which shows an explicit interplay with the Italian political and social landscape, that however seems to be disconnected from the actual geographical distribution and relocation of migrants. In addition, quite surprisingly, the influencers and political leaders that apparently lead the debate, do not necessarily belong to the clusters that include the majority of nodes: we find evidence of the existence of a 'silent majority' that is more connected to accounts who expose a more positive stance toward migrants, while leaders whose stance is negative attract apparently more attention. Finally, we see that the community structure clearly affects the diffusion of content (URLs) by identifying the presence of both local and global trends of diffusion, and that communities tend to display segregation regardless of their political and cultural background. In particular, we observe that messages that spread widely in the two largest clusters, whose most popular members are also notoriously at the opposite sides of the political spectrum, have a very low chance to get visibility into other clusters.

**Keywords:** network analysis; social media; network segregation; immigration; clusters; information cascades

## 1. Introduction

Over the last decades, millions of non-EU citizens arrived in Europe, brought by many different geo-political events [1]. Italy, being a natural bridge for the Mediterranean migration routes due to its geographical position, is particularly involved in this phenomenon, which has been at the centre of the national political and public debate since a long time [2]. Migration is frequently referred as one of the most common topics of discussion among the general population all over the world—and one of the most divisive and polarising. The intensity of the public debate might be ascribed to the ever extensive media coverage, to the social-economic challenges of migrants' integration in a complex society, and to the rising of right-wing nationalist political parties which build most of their political stances and actions around the issue of foreign immigrants [3]. The public discourse around immigration is usually a controversial one—as the New York Times was already pointing out in 2018, "*[...] It is the paradox of Europe's migration crisis: The actual number of arriving migrants is back to its pre-2015 level, even as the*

*politics of migration continue to shake the Continent. [...] The precipitous drop in migrant arrivals does not mean that Europe is without real challenges. Countries are still struggling to absorb the roughly 1.8 million sea arrivals since 2014"* [4].

Moreover, the Italian political landscape is peculiar due to its intrinsic complexity: elections and alliances are not only a matter of the internal and external dynamics within and between two main antagonistic parties (e.g., Democrats vs. Republicans in the USA); in fact, if we consider the traditional political spectrum from left to right, we have the following main parties: LeU (Liberi e Uguali—Free and Equal), PD (Partito Democratico—Democratic Party), FI (Forza Italia—Go Italy), Lega (League—formerly known as Northern League) and FdI (Fratelli d'Italia—Brothers of Italy). In addition, it must be mentioned that M5S (Movimento 5 Stelle—5 Stars Movement), that actually got the relative majority of votes in the last election held in 2018, claims to stand apart (neither a leftist, nor a rightist party, and far from the centre). In this complex scenario, migration has indeed been one of the main focus of Italian politics until 2020: during the first years of the decade the then-majority party, the leftist PD, was publicly pro-immigration, even if they produced some controversial policies, as financing a severe border control in Libya. The latest years of the decade were instead marked by the growth of both M5S and Lega, with the latter being fiercely against illegal immigration and open borders. Indeed several right-wing European political parties, especially those with an Eurosceptic stance, see border closure and security as one of the key issues in modern Europe, therefore basing an important part of their media communication on this matter [3]. Lega and M5S, in particular, formed a Government coalition that lasted from March 2018 to August 2019, having built a very large popular consent on different—and complementary—social issues [5]. M5S adopted a more opaque communicative approach about immigration, contrasting NGOs actions together with Lega and FdI, and officially supporting aids when they later formed the government with PD and LeU in 2019, which switched with Lega its place at the opposition. This quick briefing of recent Italian politics is necessary to provide the reader with context: it explains why the complexity regarding open discussion through social media on immigration is an interesting case study. In particular, supporting one party or another does not necessary mean that the voter's stance toward immigration's policies is consistent with the official position of their representatives.

Digital social media have become, since several years, a privileged venue for citizens as well as for news outlets and politicians where to debate and share opinions about topical issues. News media content, reporting on recent events and societal issues, ignite a debate among society, with people populating social media feeds with comments, not rarely using harsh tones, often assuming clear and strong stances towards specific topics. This has strong social implications on everyday life and over electoral outcomes and political opinions [6]. For this reason, digital social media might be suitable to study and analyse the public debate around specific controversial topics. In particular, we focus on the Italian public debate about the phenomenon of immigration, using Twitter as source of data, with the final goal of understanding how the debate takes place and how strong is the polarisation and the division among Italian users, as far as this topic is concerned.

This work provides an exploration of the network of all Twitter interactions around the hashtag *#migranti* (*migrants*, in Italian) during the period between August 2018 and July 2019, starting from the following working hypotheses:

1. People engaged in a divisive debate are not necessarily influenced by external factors (as already pointed out in [7]) or also 'facts' and observable events.
2. The political fragmentation and the inherent fallacies of reducing the debate on immigration to a trivial 'pro' vs. 'against' dichotomy makes polarisation in this debate more elusive than in other domains [8–12].
3. The presence of clusters with heterogeneous characterisation in the Twitter network can represent a more diversified manifestation of the so-called 'echo-chamber' phenomenon.

4. Complex and simple contagions [13–15] can concurrently trigger different opinions and news diffusion dynamics through the clusters; underestimating the 'silent majority' and focusing on hyperactive members of small communities can lead to distorted perception of what is going on.

To support the validity of the above-mentioned hypotheses, we focused on the following research questions:

- Where are the users tweeting about immigration located? We know that certain areas are more exposed to the presence of both regular and irregular migrants: regions of arrival, big cities [16], transition regions towards central Europe. Are we able to locate Twitter users posting about migrants? Is there a geographical correlation with the location of migrants, i.e., are people talking about migration where the issue is more pressing, or is it more of a national debate untied from local observable events?
- How do interactions between users with different opinions unfold? A structural study of the interaction among users discussing about the topic of migration could help us to understand the extent and dynamics of the public debate on social media. Is it really a strongly divisive topic? Who are the actors driving the debate, and are we able to identify factions with different stances? Is the size of such clusters somehow proportional to the popularity of their leaders?
- What is the role of clusters in the diffusion of opinions and news? Given that the migration debate network on Twitter is divided in many clusters, we wonder how opinions and news shared within those groups of accounts spread to other communities. Do locally popular news have the chance to get also distant clusters? Do a cluster play a role as a barrier or an accelerator of diffusion?

## 1.1. Paper's Structure Road Map

The present work is structured as follows: after reviewing the relevant scientific literature, we will briefly present our data in Section 2; in Section 3, we will review the methods used to geolocate the tweets and to build and analyse the networks, as well as to study the polarisation and the diffusion of content in the community-induced graphs, and we will present the results. Finally, in Section 4, we will discuss these results, also in light of the existing literature, and in Section 5, we will summarise and point out some unanswered questions that will build the foundations for future work.

## 1.2. Related Work

Extensive research work has been carried out in a vast number of different fields, about both migration and all the debate around it. It has been highlighted how, during the recent years, the tone of the debate has radically changed. As far as Italy is concerned, McMahon analysed all the speeches on migration given in the Italian parliament over the period from May 2008 to April 2011 [17]. He found that, despite high migration flows, rising high unemployment, aggressive government austerity and economic uncertainty, there was a low level of salience of immigration in public opinion but at the same time there was a collapse of right-wing political parties amidst scandals and increasing distrust from the public [17]. This situation has now been reversed, with right-wing political parties on the rise while left-wing parties are currently facing a deep loss of consensus, in Italy as in most of Europe. According to the report about the public opinion on migration published by the Migration Data Portal (managed by the International Organisation for Migration), some evidence suggests it is politics that drives public debate on migration rather than the actual numbers characterising the phenomenon; they analysed multiple data sources to conclude that, as a general trend, public opinion is strongly divided on the issue of increasing, decreasing or keeping present immigration levels, with a slightly higher percentage of people supporting the "decrease" option [18]. This is a complex phenomenon, and the relationship between public debate, news media and policy making is not uni-directional: statistical evidence has been found that the sentiment in public debate is strongly predictive of European asylum acceptance rate and, in general, strongly correlated with asylum policies of European Countries [19].

When analysing public discourse in social media, it is important to take into account the role of homophily, selection and influence in shaping the dynamics of the debate. They are particularly relevant in the context of political networks on Twitter, that are found to be highly segregated as users tend to retweet a lot more content generated by users with the same political orientation [8,9]. In [7], authors show how big polarised communities, like those of Trump's and Clinton's supporters, tend to remain separate and to ignore the influence of live events, such as topics debated during a TV confrontation; a similar analysis on the influence of external factors on the Twitter debate was carried out for the Brexit event [20].

Similar conclusions about polarisation can be drawn in the specific context of political conversations in Italian on Twitter [11]. Homophily can sometimes be amplified—or at least be very evident—on social networks. Its effect might vary, depending on the political culture and background of users [21]. This is not limited to politics; homophily and the tendency to interact only with people with similar opinions are mechanisms underlying the nature of human interaction. The phenomenon of echo chambers has been recently studied in another divisive topic: the Italian Twitter debate on vaccinations [12]. It has been found how marked is the presence of echo chambers in both pro-vax and anti-vax communities, even with different topologies, most likely due to different cultural and social backgrounds of the communities. It is important to observe that we can reason about the diffusion of content and ideas in networks, as well as the role played by the community structure, on different layers of complexity. In [22], by studying the online competition between anti-vax and pro-vax during the 2019 measles outbreak, the authors put an accent on the role of the undecided, i.e., those that cannot be traced back to a community with a defined stance, and on their proximity to the anti- and pro-vax echo chambers. However, vaccination is not the only divisive topics where the so called "silent majority" that do not take explicitly take one side has a not neglectable impact; in fact, bridges and weak ties can play a crucial part in the diffusion, especially in cases such as the object of our study, where social reinforcement of ideas might be involved and dynamics of complex contagion might be taken into account [13–15]. The effect of closed discussion communities is typical not only of Twitter, but of the blogosphere in general, especially when it comes to political-related issues [10].

With respect to research works on the migration phenomenon, Twitter has been used to study the level of social mixing within big cities, to understand how the so-called global cities integrate the diversity of languages and cultures of migrants [16]; migrations or, more in general, short and long term global mobility patterns, have been explored also through Twitter data [23–26]. The Twitter discourse on migration has been studied with different methods and scopes in several national contexts, e.g., in Russia [27] or Germany [28]. The Italian Twitter debate on migration has been studied with the specific goal of automatically detect hate speech towards migrants [29].

Our novel contribution to this general context can be summarised as follows: we collected and explored a Twitter dataset of tweets around the topic of migrations, covering a particular time window (August 2018–July 2019) when this was among the most debated topics in Italy, as it was one of the main concerns of the Government. On one hand, we map the geographical location of users tweeting about migration to assess whether the fact of living in an area affected more by immigrants influxes might influence the volume and content of Twitter posts. Moreover, by analysing the network of interactions among Twitter users, we try to understand to what extent network homophily and high visibility of some popular user affect communication and the diffusion of content, by exploring different features of the debate like URL sharing and word usage, in light of the properties of the network.

## 2. Data

*Twitter Data Collection and Pre-Processing*

With about 330 million monthly active users, Twitter is one of the most popular micro-blogging platform in the world. Each user can post their tweets, having available up to 280 characters to express their thoughts or feelings about a particular topic, which is frequently accompanied by hashtags. Users can also engage in conversations by replying to other users' tweets, retweeting them, i.e., sharing their tweets, or quoting them, i.e., retweeting and adding a personal comment or even by mentioning them in a tweet. The terminology we will use throughout the whole paper is summarised in Table 1.

**Table 1.** Terminology used throughout the paper.

| Tweet | Meaning |
|---|---|
| Tweet | Original content posted by a user. A tweet can also include a mention to other users, or an URL pointing to an external website. |
| Mention | The act of mentioning another user, by citing their nickname after the '@' symbol. |
| Retweet (RT) | Another user's post, shared by the author of the retweet. |
| Quote | A retweet with an additional comment by the author of the retweet. |
| Reply | The act of replying to another user by clicking on "*Reply*" under the original tweet. The addressed user is mentioned with the '@' symbol, at the beginning of the reply message. |

We focus on the Italian debate about migration/migrants on Twitter, a platform that in Italy has a rather good uptake. Even though it does not reach the numbers of Facebook it still has a good penetration rate, with more than 2 millions users that, according to Twitter, could be reached with adverts in January 2019 [30]. Moreover, a significant correlation (0.93) has been found between the number of (geotagged) tweets in Italian and the total resident population in each Italian province [31]. Thus, we used Twitter's Stream APIs to collect tweets in Italian, filtering on the keywords **migranti, immigrati, immigrazione** (in English: migrants, immigration). We chose these particular keywords as a filter for our data collection as they were the most neutral ones in the debate. There were others we could have chosen (e.g., *clandestini* or *rifugiati*, respectively *illegal immigrant* and *refugee*) but we chose not to, as they are particularly used either against or pro-migration factions and could introduce unwanted bias in our data.

The phase of data collection started in August 2018 until July 2019, collecting almost 6 million tweets produced by 301,664 unique users. A detailed view of the composition of the dataset can be found in Table 2. Data were then parsed, cleaned and stored.

**Table 2.** Composition of the Twitter dataset. Retweets are the vast majority.

| Composition of the Twitter Dataset | |
|---|---|
| Retweets | 4,424,359 |
| Quotes | 1,029,317 |
| Replies | 402,177 |

To assess the representativeness of our dataset, we compared our volume of traffic with the volume of queries on Google about the term **migranti**. The trends, that can be seen in Figure 1, look highly comparable, with peaks only slightly shifted, most likely due to the different time availability of the data (Twitter data have a daily record, while Google Trend data are related to a 1-week time window). Each peak corresponds to a major event happened in Italy during the period of observation and related to migrants. Please see the caption in Figure 1 for a detailed list of events.

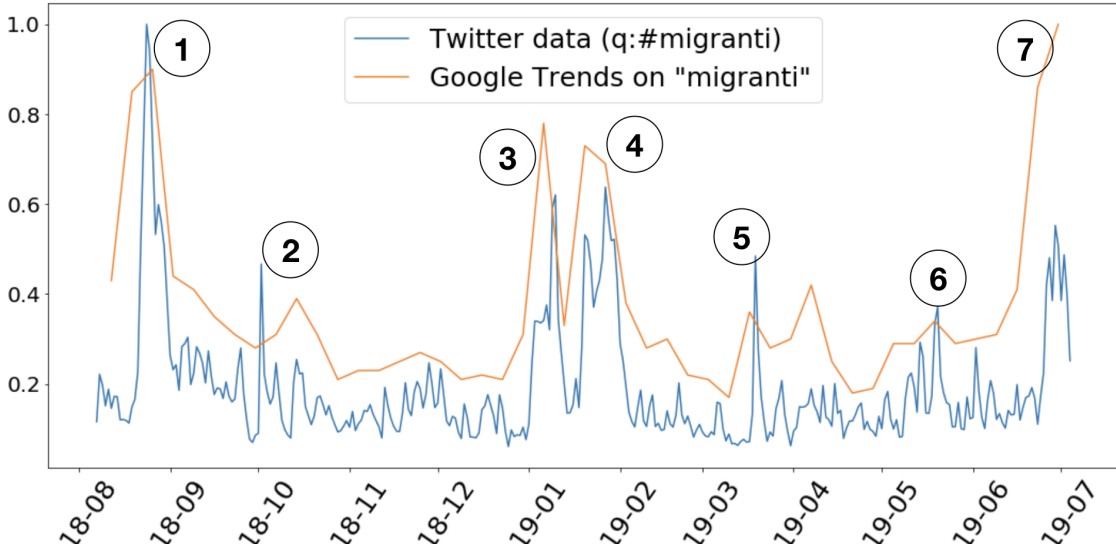

**Figure 1.** Comparison between Google Trends on '#migranti' (orange line) and volume of tweets filtered on the same keyword. The more irregular pattern in the blue line is probably due to the much higher availability of Twitter data, that helps in providing a higher resolution picture of the trend. Google queries data was available on a weekly basis only. The spikes in the data are due to the following events: (1) a NGO ship held outside of a harbour for a long time and finally let dock (August 2018), (2) the arrest of Mimmo Lucano, a City Mayor accused of favouring illegal immigration (October 2018), (3–4) a minor arrival highly covered by the news media and the International Memorial Day (January 2019), (5) seizure of the Mare Jonio NGO ship (March 2019), (6) intense month of arrivals on the Italian coasts (May 2019) and (7) arrest of the NGO ship Captain Carola Rackete.

## 3. Methods and Results

### 3.1. Where Are the Users Located? Geo-Spatial Analysis

It might be useful, in order to gain a better understanding of the nature of the debate, to locate the users tweeting about migration and to cross-check this information against the distribution of migrants over the Italian territory. This could possibly tell us if the actual presence of migrants plays a role in the ignition of the debate; thus, if the public discourse on social media is linked with territorial issues rather than being a national matter.

#### 3.1.1. Geolocation of Tweets

Explicitly geolocated tweets were less than 1% of the total number of collected tweets. Therefore, since a geographical information on the tweet could be a great addition to our analysis, we simply retrieved this information also from other sources, to assign an alleged location to the tweet. Each tweet downloaded with the Stream API contains several information not only about the tweet itself but also about the user profile. We decided to assign a location to the tweet following this hierarchy of progressively less precise information:

- If geotagged information is present (coordinates detected by the Twitter client, if access to the geolocation of the device was granted by the user), this will be the preferred information.
- Otherwise, if the user checked-in a place when posting the tweet, this information will be used.
- Finally, we would look if the user set a home location in his/her profile.

If any of the three above information are present, the most precise one would then be selected as geolocation for the tweet. Names of the locations were checked against the official Italian National Institute for Statistics (ISTAT) nomenclature for Italian regions, provinces and municipalities. After this phase, about 1.6M tweets, roughly the 26% of the dataset, were labelled with a location. We recall that in [31] a correlation of 0.93 was found between the number of geotagged tweets and residents per province. In our case, we reach a value of 0.86 (with a *p*-value of $9.9 \times 10^{-33}$), which is slightly lower but still high enough to corroborate the validity of our geolocation pipeline.

### 3.1.2. Correlation with Institutional Spatial Data

We now correlate Twitter locations with official data, gathered from *Immigrati.Stat* (http://stra-dati.istat.it/), the ISTAT online archive specialised on data about migrants and foreigners in Italy. All data were available at different levels of granularity, corresponding to the official administrative levels (regions, provinces and municipalities) and were usually detailed per gender, age cohort, country of origin. Studying this data can also help in putting into context the whole situation about migrants in Italy. From the website of the Ministry of Interior, we learned that the number of migrants illegally landed in Italy by boat in 2019 (until the 4 October) is <8000, following a strongly decreasing trend that already started in 2017 (precisely: 108,876 in 2017, 21,119 in 2018 and 7896 to 4 October 2019). The distribution of the total number of residence permits, normalised over the number of residents in the area, can be seen in Figure 2. From this map, it is clearly visible how newcomers try to settle mostly in northern Italy, with particular hotspots in the province of Milan, Rome and in different provinces between Emilia-Romagna, Tuscany and Lombardy.

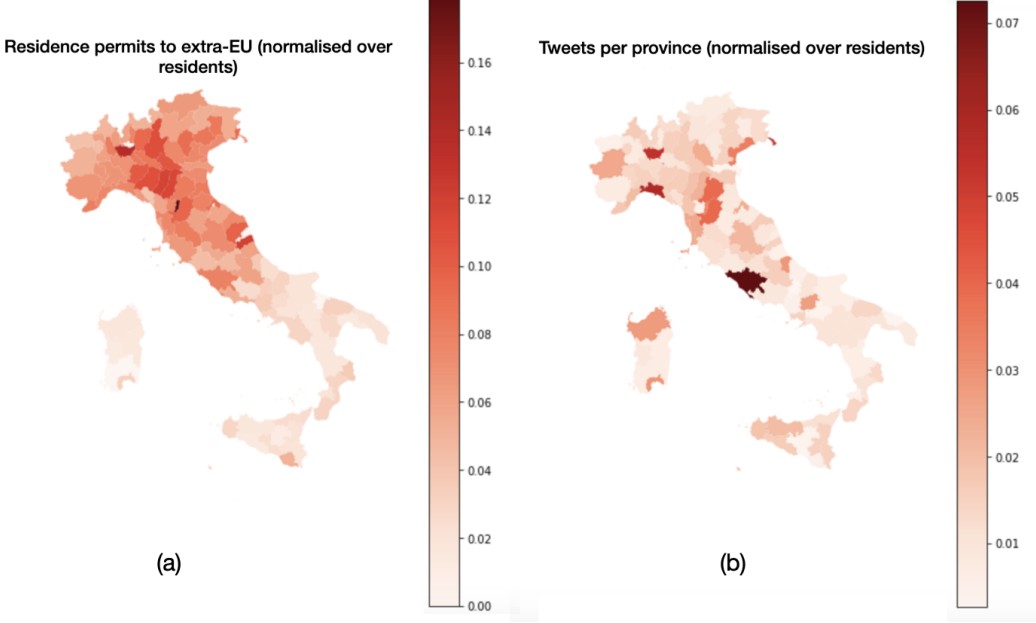

**Figure 2.** Spatial distribution of the geolocated tweets (**b**) compared to the distribution of residence permits per province (**a**). Both are normalised over the number of residents per province. We can observe that the provinces that display the highest activity are Rome, Milan and Genoa. It is also clearly visible the gap in terms of Twitter activity—at least on this specific topic—between northern and southern Italy. The correlation with residence permits appears to be low (Pearson's correlation coefficient = 0.37, with *p*-value = $8.08 \times 10^{-5}$).

Keeping in mind that all data about the territorial distribution refers to those migrants whose situation is somehow regularised and acknowledged by the State (i.e., irregular migrants without residence permits are not included), we can see in Figure 2 that the correlation between number of tweets and resident permits looks quite low, maybe with the only exception of the province of Milan. Pearson's correlation confirms this sensation, with data being only mildly positively correlated $(0.37, p\text{-value} = 8.08 \times 10^{-5})$. Therefore, the debate seems to be driven by other mechanisms rather than the actual, local presence of migrants.

### 3.2. How Do They Interact? Analysis of the RT Network

In order to understand how interactions and communications unfold among users, we adopt a network analysis approach. We can build at least three different networks, one for each type of Twitter interaction: a retweet (RT) network (which includes quotes), a reply_to network and a mention network. In particular:

- **RT network**: each node is a Twitter user, and a link is established between two nodes if one of the users retweets or quotes the other. The link is weighted, with the weight representing the number of RTs and quotes between the two users.
- **Reply network**: each node is a Twitter user, and a link is established between two nodes if one of the users replies to the other. Here as well, the weight is the number of replies between the two.
- **Mention network**: each node is a Twitter user, and a link is established between two nodes if one of the users mentions the other. The same weighting mechanism applies to this network.

Each network is weighted based on the number of repeated interactions between the same pair of nodes. Other basic statistics on the networks are displayed in Table 3.

**Table 3.** Basic statistics of the migration debate Twitter networks.

|  | **RT Network** | **Reply Network** | **Mention Network** |
|---|---|---|---|
| **Number of nodes** | 232,384 | 75,319 | 130,202 |
| **Number of links** | 2,210,242 | 280,318 | 770,505 |
| **Density** | $8 \times 10^{-5}$ | $1 \times 10^{-4}$ | $9 \times 10^{-4}$ |
| **Size of Giant Component** | 227,597 | 75,319 | 130,202 |
| **Modularity** | 0.48 | 0.44 | 0.25 |

For the purpose of the community detection, since we are using the Louvain algorithm, we consider the networks as undirected. For other measures, we will keep the directions into account.

The present work is focused on the RT (Retweet) network, since retweets are the vast majority of events in our dataset. In addition, we want to focus on structural characteristics that are shaped according to a clear homophily principle, in order to detect graph clusters made of users that are more likely to share same viewpoint. Empirical studies have proven that a retweet between two users is more likely to be interpreted as endorsement [8,9,11], whereas 'reply to' or 'mention' can also manifest disagreement. Particularly, we concentrate on the largest connected component in the RT network which contains 97.9% of the nodes.

In order to assess the presence of assortativity in the RT network, we compute the average nearest neighbour degree [32], whose results are shown in Figure 3a. We find a slightly disassortive behaviour further confirmed by the comparison with a null model obtained with a degree-preserving randomisation of our network through a configuration model (see Figure 3b). Hence, we have that lower degree nodes (i.e., ordinary citizens) are very likely to retweet messages produced by high-degree nodes (i.e., account associated to very popular persons).

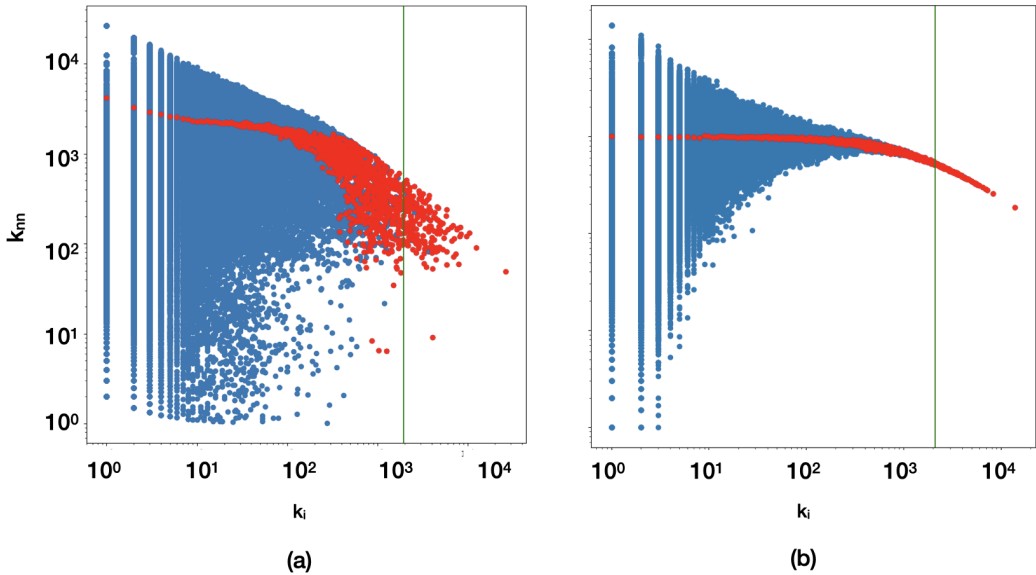

**Figure 3.** Comparison of the $k_{nn}$ of the retweet (RT) network (**a**) with a configuration model (**b**). The green line corresponds to the structural cutoff value for the degree, while the red dots represent the mean value for each "class" of nodes of degree $k$. The network displays a slightly disassortative behaviour.

By ranking the nodes according to their degree centrality, we find that the most central nodes with respect to the degree can all be traced back to news media, journalists and politicians (Table 4). This is not entirely unexpected, given the very nature of the system under examination; the identity and the role of opinion leaders in political debates on Twitter has already been studied in scientific literature [33]. More in general, the nodes with the highest degree, i.e., those driving the debate, represent those same public actors that might fuel the offline debate, while the "common citizens" remain more on the sidelines, often limiting themselves to a passive engagement in content sharing rather then being proactive.

**Table 4.** Table describing the communities of the RT network in terms of sizes, density, number of posts with hashtag "migranti", top 10 nodes for in-degree centrality and top 10 most used hashtags (excluding #migranti).

| ID | Size | Internal Link Density | No. of #Migranti | Highest In-Degree Nodes (Usernames) | Top Hashtags |
|---|---|---|---|---|---|
| RT1 | 116,831 | $1.5 \times 10^{-3}$ | 51,639 | Gad Lerner, Roberto Saviano, La Repubblica, Linkiesta, Udo Gümpel, Fabio Niccolò Zancan, jacopo iacoboni, Nello Scavo, laura boldrini. | #salvini, #diciotti, #facciamorete, #immigrazione, #seawatch, #pd, #ong, #italia, #riace, #restiamoumani |
| RT2 | 34,174 | $1.93 \times 10^{-2}$ | 62,980 | Giorgia Meloni, Cesare Sacchetti, Francesca Totolo, Diego Fusaro, La Verità, ImolaOggi, Giank-deR, Claudio Perconte, Il Sofista, Antonio M. Rinaldi . | #salvini, #immigrazione, #diciotti, #seawatch, #libia, #facciamorete, #riace #lega, #portichiusi, #salvininonmollare |
| RT3 | 27,845 | $2.4 \times 10^{-3}$ | 4575 | Matteo Salvini, Lega - M. Salvini Premier, Noi con Salvini, TG2, Attilio Fontana, Generazione Identitaria, Marco Morini, Cittadina Italiana, Don Alphonso, Matteo SALVINI | #salvini, #diciotti, #immigrazione, #pd, #libia, #italia, #ong, #decretosalvini, #salvininonmollare, #portichiusi |
| RT4 | 9553 | $3.5 \times 10^{-3}$ | 8887 | Il Fatto Quotidiano, Danilo Toninelli, Peter Gomez, Carlo Sibilia, Movimento 5 Stelle, Franco Bechis, Andrea Franchini, Le Frasi di Osho, Elio Lannutti. | #salvini, #diciotti, #immigrazione, #pd, #facciamorete, #m5s, #seawatch, #ong, #lega, #portichiusi, #dimaio |
| RT5 | 9225 | $2.4 \times 10^{-2}$ | 6193 | SkyTg24, ANSA, Tgcom24, RaiNews, Agorà Estate, Agi Agenzia Italia, Adkronos, Dagospia, Ultime Notizie, Il Messaggero. | #salvini, #diciotti, #immigrazione, #facciamorete, #allnews24, #seawatch, #dimaio, #AGI, #nonstopnews, #libia |

### 3.2.1. Community Structure and Analysis of Content: Polarisation of the Debate

In order to capture the polarisation of the debate, we investigate the community structure of the network. Empirical evidence tells us that retweet (and quotes, to a slightly lower extent) can be seen as endorsement [11], especially when the conversation happens in restricted communities rather than across a non-cohesive network [34]. Therefore, we check the most internally cohesive sub-groups of the RT network to find those nodes who are most likely to share the same stance towards the topic of migration.

We use the well-known Louvain method for community detection, an agglomerative algorithm that merges together groups of nodes to maximise modularity [35]. In our case, the best partition has a modularity of 0.48. It is interesting to look at the composition of the communities. As we can see in Figure 4, there is a high number of micro-communities composed of only two or three nodes.

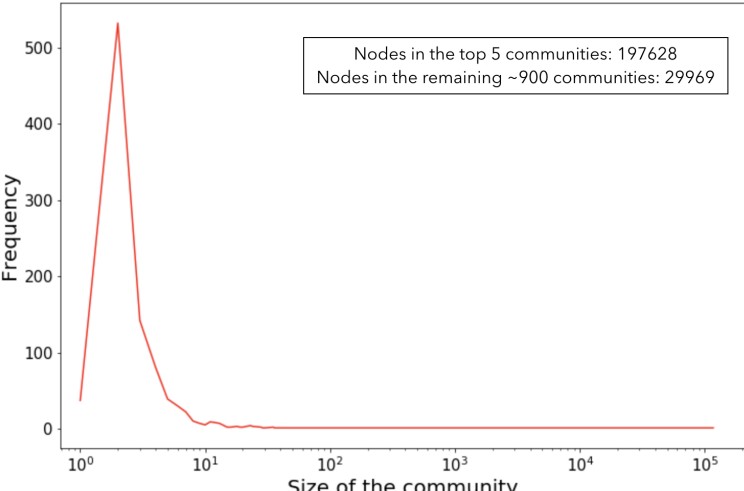

**Figure 4.** Size of the communities. There is a great number of micro-communities made of only two or three nodes. The first 5 communities group together 87% of the nodes, leaving out only 29,969 nodes.

We focus on those groups who present at least 1000 nodes and with more than 100 RTs that could be traced back to their members. This results in the five communities described succinctly in Table 4. Moreover, by looking at the top 10 most highest in-degree nodes in each community, we can provide a conceptual annotation of the various groups. Their in-degree distribution can be seen in Figure 5.

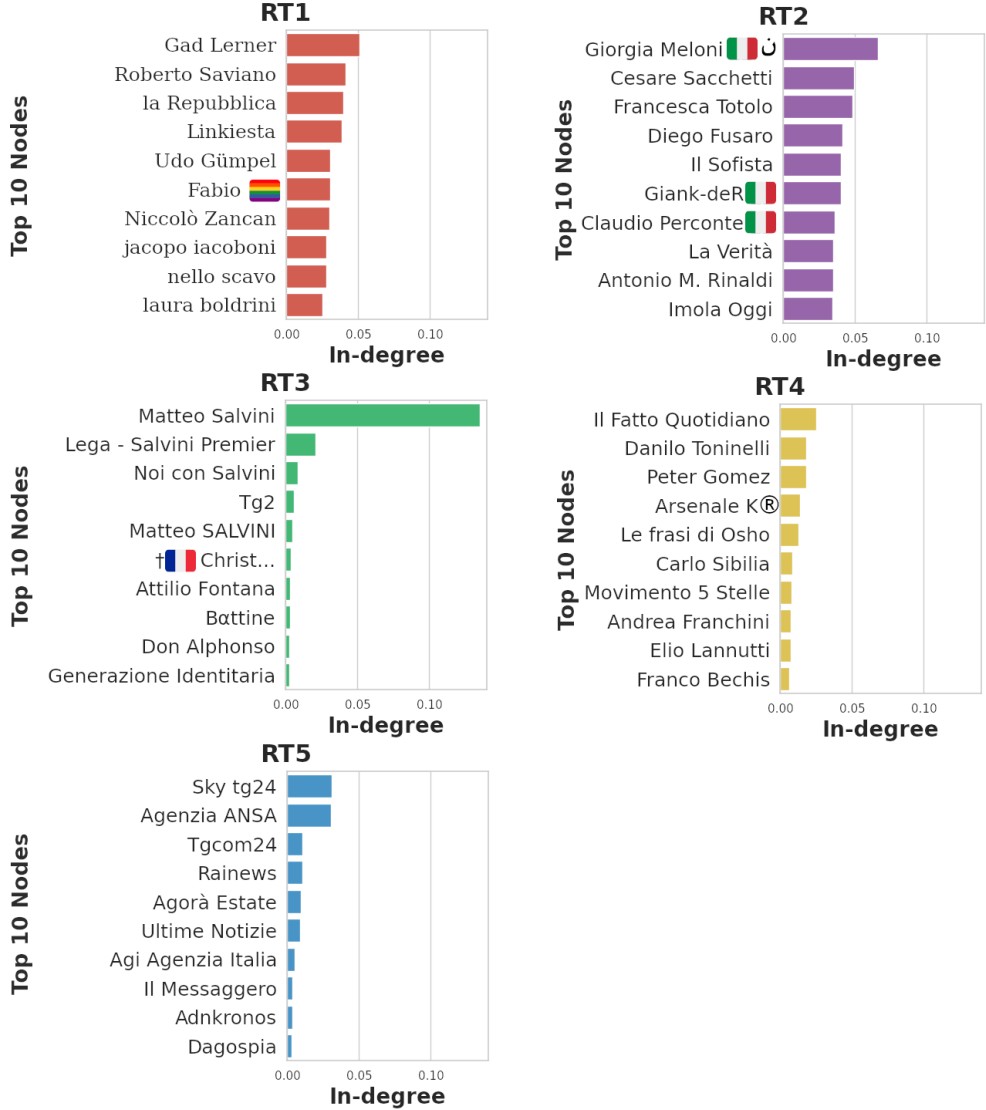

**Figure 5.** In-degree distribution of the top 10 nodes in each community. The degree centrality values are normalised by dividing by the maximum possible degree. We can see how a relatively smaller community, like RT3, sets the scale for all the others thanks to the presence of Matteo Salvini, one of the most active Italian politicians on social media.

The biggest community, that includes more than 110,000 nodes, has among its most representative users many journalists (Gad Lerner, Roberto Saviano, Udo Gumpel, Niccolò Zancan, Iacopo Iacoboni, Nello Scavo), two newspapers (La Repubblica, Linkiesta) and a politician (Laura Boldrini), whose stances toward the topic are generally recognisable to be positive, in favour of welcoming migrants in need. The other communities are significantly smaller but still very much involved in the debate. Community RT2 has indeed the highest number of posts with the hashtag #migranti (62,980 posts), even though it is more than three times smaller than RT1 (34,174 accounts in RT2, and 116,831 in RT1). The highest in-degree nodes of this community correspond to people with a completely opposed view with respect to principal members of RT1, being politicians, newspapers and celebrities who are publicly and vocally against foreign immigrants, often in close liaison with nationalist and right-wing parties, such as Giorgia Meloni, the extreme-right FdI party leader. Quite interestingly, RT3 (27,845 accounts, 4575 posts) and RT4 (9553 accounts, 8887 posts) can be linked to the two government parties, such as, respectively, Lega and M5S. In fact, we can easily find the Lega's leader (Matteo Salvini) official account, as well as some supporters (e.g., Lega—M. Salvini

Premier, Noi con Salvini, matteo SALVINI), or other Lega's politicians (e.g, Attilio Fontana) in RT3; also, in RT4 we find some famous M5S politicians (e.g., Danilo Toninelli, Carlo Sibilia, Elio Lanutti), the official M5S party account, some journalists (e.g., Peter Gomez, Franco Bechis) and a newspaper (il Fatto Quotidiano) that covered extensively the government internal debate on this topic. Finally, RT5's most retweeted accounts are linked to news outlets (SkyTg24, ANSA, Tgcom24, RaiNews, Agorà Estate, Agi Agenzia Italia, Adkronos, Dagospia, Ultime Notizie, il Messaggero).

If we take a look at the in-degree distributions, we find something even more interesting. Matteo Salvini's account (RT3) is by far the most retweeted in the migration debate, followed by Giorgia Meloni (RT2) and Gad Lerner (RT1). In addition, considering RT2's smaller sizes with respect to RT1, it should be noted how its 10 top ranked nodes are retweeted within the same order of magnitude of RT1's 10 highest in-degree accounts. It is striking that Salvini and RT2's highest in-degree user are driving the discussion on this topic more efficiently than most popular RT1's members do. Therefore, we have one very large community that is formed around users whose stance is officially positive towards migrants and refugees, and two smaller communities whose members are very much engaged by retweeting posts created by their own influencers. These apparently contradictory results can be interpreted by admitting that there is a 'silent majority' of users that retweet occasionally posts created by RT1's influencers, while 'common users' in RT2 and RT3 dedicate much of their efforts to spread tweets created by the leaders of Lega, FdI and other people whose stance toward illegal migration is quite negative.

To better assess clusters' cohesion, we calculated the internal link density as:

$$d_c = \frac{2L_c}{N_c(N_c - 1)} \tag{1}$$

where $L_c$ and $N_c$ are respectively the number of internal links and nodes in each community $c$. These values are displayed in the third column of Table 4. Again, we can observe that, despite all the 5 clusters show high cohesion compared to the whole network that has density $8 \times 10^{-5}$ (see Figure 3), RT2 shows to be the denser community ($1.93 \times 10^{-2}$), stressing out that members of this subgraph are more cohesive and more engaged.

We can finally shine more light on the relationships between communities taking into account the directions of incoming and outgoing links. The heatmap in Figure 6 represents the weighted adjacency matrix of the community-induced graph, normalised by row. We can see that, for both RT3 and RT4 (the communities mainly linked to the two Government parties League and 5 Stars Movement), 84% of the outgoing edges are towards RT2 (i.e., accounts in RT3 and RT4 retweeting RT2's members posts) which, from what we can gather from our analysis, represents the group of accounts that strongly oppose immigration, and that in general share the same right-leaning ideology. At the same time, RT5 (News Media) and RT1 (Democrats and, more in general, pro-immigration users) are strongly connected to each other, both sharing ∼50% of their total retweets. In general, by looking at the well-known accounts corresponding to the 10 highest in-degree nodes for each community, we can observe that users seem to be clustered in a coherent fashion.

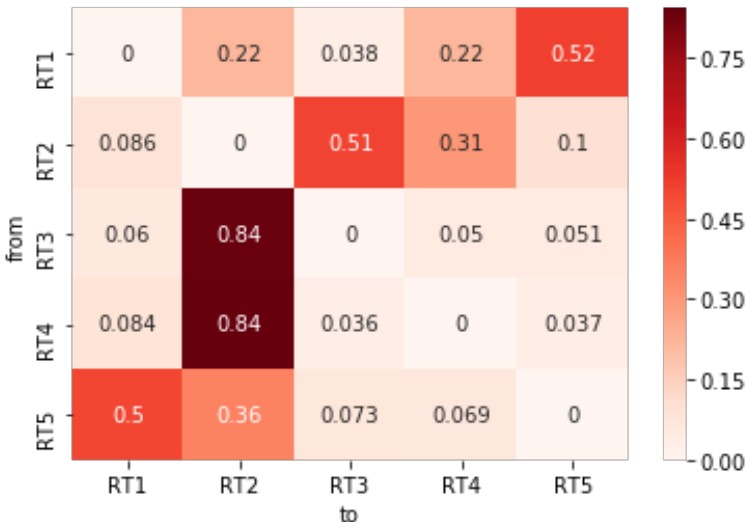

**Figure 6.** Weighted adjacency matrix of the community-induced directed RT network, normalised by row.

Figure 7 gives us a visual representation of the connections between the communities from quantitative information extracted from Figure 6.

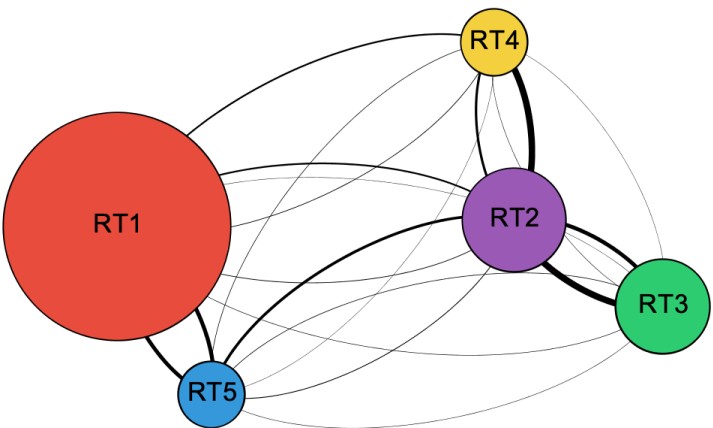

**Figure 7.** The graph induced by the community detection on the RT directed network, whose adjacency matrix is given in Figure 6. Each super node represents a community: sizes are proportional to the number of accounts in each community. Links' directions are given clockwise.

### 3.2.2. Characterisation of the Communities: Hashtags and N-Grams

Characterising the clusters simply by looking at the highest degree nodes does not allow us to say much about the remaining members of the communities: we are basing our analysis on the empirical evidence that suggests retweet is endorsement—and, to a slightly lower extent, quotes are as well [11]—but we want to further check if this holds, without simplistically having the users "inheriting" the stance of the central nodes of the community they belong to. Therefore, we inspect all hashtags and n-grams produced by every user, who is labelled with the community they belong to in the RT network, finally calculating a mean result per community. This would allow us to better understand if the users share the same stance of the central nodes of their communities, whose stance is known and used to "mark" the clusters.

The analysis of the hashtags does not show significant differences (see Table 4). The hashtags used across the various communities seem to be rather similar, regardless of the stance of the central nodes:

the name of Salvini, the League's leader and, back then, Minister of the Interior, is always the most common hashtag. Only towards the end of the ranking could we find some hashtag that might be more peculiar and relevant to the community, like #portichiusi (closed harbours) and #salvininonmollare (Salvini do not give up) for RT2 and RT3, or #restiamoumani (stay human) for RT1.

The analysis of n-grams, and bi-grams in particular, is probably more interesting. N-grams are sequences of $n$ words; we analysed n-grams for $n = 1, 2, 3$ to check respectively the most used words and the most used combinations of 2 and 3 words. To prepare the text for the analysis, we applied a standard textual data cleaning pipeline, which includes removal of typical Twitter items like hashtags, mentions and "RT" characters, removal of punctuation and of stop words after tokenisation. We did this analysis for every tweet of every user; here, we present the results aggregated by community. We consider the 5 most relevant communities for our analysis (RT1-RT2-RT3-RT4-RT5).

We identified a certain number of bi-grams that are common to all communities (Table 5), with the only exception of RT5, and other bi-grams that instead can be found only in some communities, while are absent from others (Table 6). The complete distribution of the 20 top bi-grams can be seen in Figure 8. Table 6 helps us understanding the stance of the communities towards migrants. First, we can clarify the unexpected results for RT5: all bi-grams refer to a quote of the Minister of Interior of Luxembourg, who used the French expression Merde alors while replying to the Italian Minister of Interior Matteo Salvini, who improperly considered it as an insult, causing a huge fuss among Italian news media. We can make more concrete claims about the other communities: RT1 uses terms like "safe harbours", "human rights" or "human beings" that appeal to a more sympathetic stance rather than RT2-3-4, who all refer to aiding and abetting illegal immigration, uncontrolled immigration, economic migrants (as opposed to migrants in need who are escaping from Countries at war: most African migrants are considered economic migrants by Italian right-wing parties). This result strongly supports the characterisation that we carried out by looking at the highest degree nodes only.

**Table 5.** Bi-grams in common to all communities, with the only exception being RT5, which did not share any bi-gram with the other communities.

| ID | Bi-Grams (IT) | Bi-Grams (EN) |
|---|---|---|
| Common to all communities but RT5 | Sea Watch, Ministro Interno, Guardia Costiera, Immigrazione Clandestina, Migranti Italia, Migranti Salvini | Sea Watch, Ministry Interior, Coast Guard, Illegal Immigration, Migrants Italy, Migrants Salvini |

**Table 6.** Bi-grams characterising the communities. There is a clear alignment between communities RT2-3-4. RT5 stands out, mostly quoting a speech from the Ministry of Interior of Luxembourg. RT1 looks like the only cluster with a positive stance towards migrants.

| ID | Bi-Grams (IT) | Bi-Grams (EN) |
|---|---|---|
| RT1 | Porto Sicuro, Porti Chiusi, Esseri Umani, Diritti Umani | Safe Harbor, Closed Harbors, Human Beings, Human Rights |
| RT2 | Favoreggiamento Immigrazione, Immigrazione Incontrollata, Migranti Economici, Migranti Clandestini, Esseri Umani | Abetment Immigration, Uncontrolled Immigration, Economic Migrants, Illegal Migrants, Human Beings |
| RT3 | Favoreggiamento Immigrazione, Immigrazione Incontrollata, Migranti Economici | Abetment Immigration, Uncontrolled Immigration, Economic Migrants, Illegal Migrants |
| RT4 | Favoreggiamento Immigrazione, Immigrazione Incontrollata, Migranti Economici, Fatto Quotidiano | Abetment Immigration, Uncontrolled Immigration, Economic Migrants, Illegal Migrants, *Fatto Quotidiano* (Italian newspaper) |
| RT5 | *Merde Alors*, Migliaia Italiani, Colorita *Merde*, Italiani Venuti, Conclude Espressione, Pazienza Sbotta, Lussemburgo Caro | *Merde Alors* (French), Thousands Italians, Colorful *Merde*, Italians Came, Ends Expression, Patience Snaps, Luxembourg Dear |

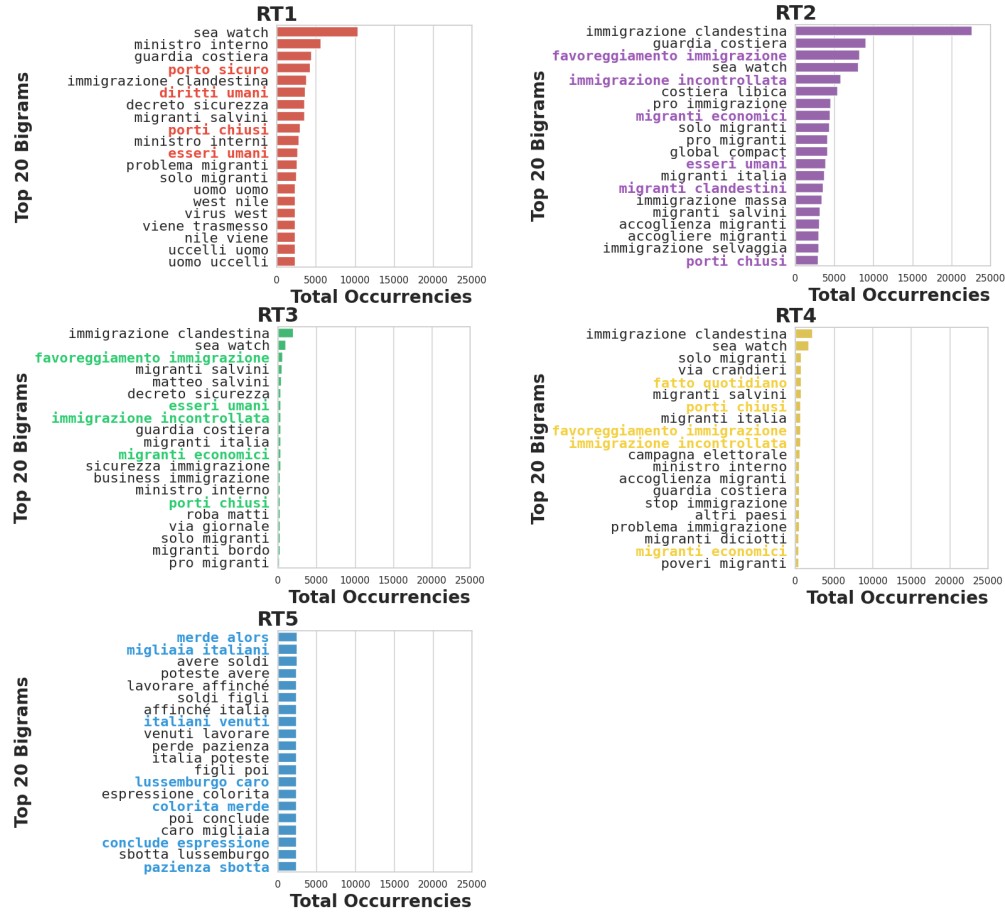

**Figure 8.** Distribution of the bi-grams among the communities. The common scale for the x-axis highlights the heavy usage of the pair illegal-immigration by RT2 users.

### 3.3. Quantifying the Diversity among Communities: Analysis of Shared URLs

Along with user's original textual content, tweets can contain links to external resources (URLs) that users want to share. We try to quantify the differences between communities by defining their diversity in the URLs shared by users (for general statistics and for an overview of the domain names of the most shared URLs in the network, please see Appendix A).

To do so, we select all the URLs with at least 100 shares and we look at the distribution of their shares among the communities. To each URL we can associate a sequence of communities sharing it over time. Let us suppose that a generic $URL_i$ is shared by users following the pattern in Figure 9. We will then assign to $URL_i$ a sequence of communities in the form of a vector:

$$URL_i \longleftarrow [RT1, RT2, RT4, RT5, RT2] \qquad (2)$$

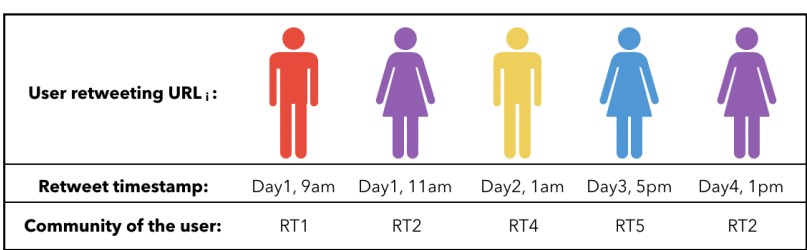

**Figure 9.** Example of sequence of $URL_i$ shares.

We are thus able to create, starting from this vector and keeping the timestamps into account, a community-wise sharing pattern for every URL. In Figure 10 is shown such a pattern for a randomly chosen URL.

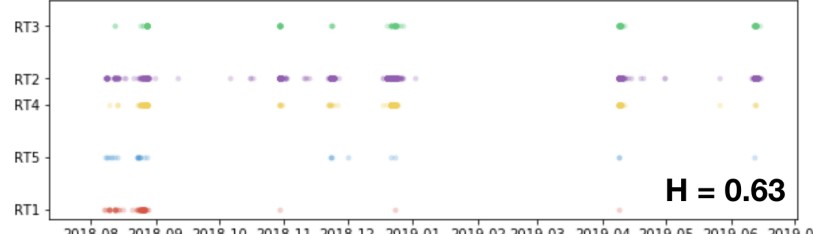

**Figure 10.** Shares over time, grouped by community, of a randomly chosen URL. The entropy for this specific sharing pattern is H = 0.63.

Inspired by [36], in order to quantify the diversity in communities sharing an URL, i.e., how heterogeneous is the reach of a URL, we can compute the entropy $H(URL_i)$ of the associated vector

$$H(URL_i) = - \sum_{c \in C} u_c \ln(u_c) \qquad (3)$$

where $u_c$ is the fraction of shares by community $c$ over the total. This enables us to quantify the heterogeneity of allocation of an URL across communities, rather than focusing on the dominant community for each URL. The higher the entropy, the more communities are sharing that URL.

Now that every URL is assigned its own entropy, we can finally characterise the communities by the entropy distribution of all the URLs they shared (Figure 11). The result is peculiar: RT1 and RT2 share a very high number of low-entropy URLs (with a median of 0.00 and 0.34, respectively), while the distributions for RT3-4-5 are much broader and heterogeneous, with much higher medians and mean values. To interpret this result we have to go back to the composition of the communities (Table 4): RT5 is a community of national news media and blogs, RT3 and RT4 are linked with the two parties of the then-Government coalition, respectively the League and the 5 Stars Movement, while RT1 and RT2—those with the lowest entropy—can be traced back to newspapers and politicians of the opposition party (Partito Democratico, leftist area) and to far right politicians and hoax spreader news media, respectively. This result establishes an interesting link between the attitudes of two areas that are at the odds with respect to their stance towards the topic of immigration. Their stance and their cultural background are different, yet they both seem to be tightly folded in on themselves, sharing URLs that can hardly ever be found in other communities. Furthermore, it suggests that there are clear global trends that spread through the network regardless of the community structure, as well as local trends that find boundaries on the edges of the clusters.

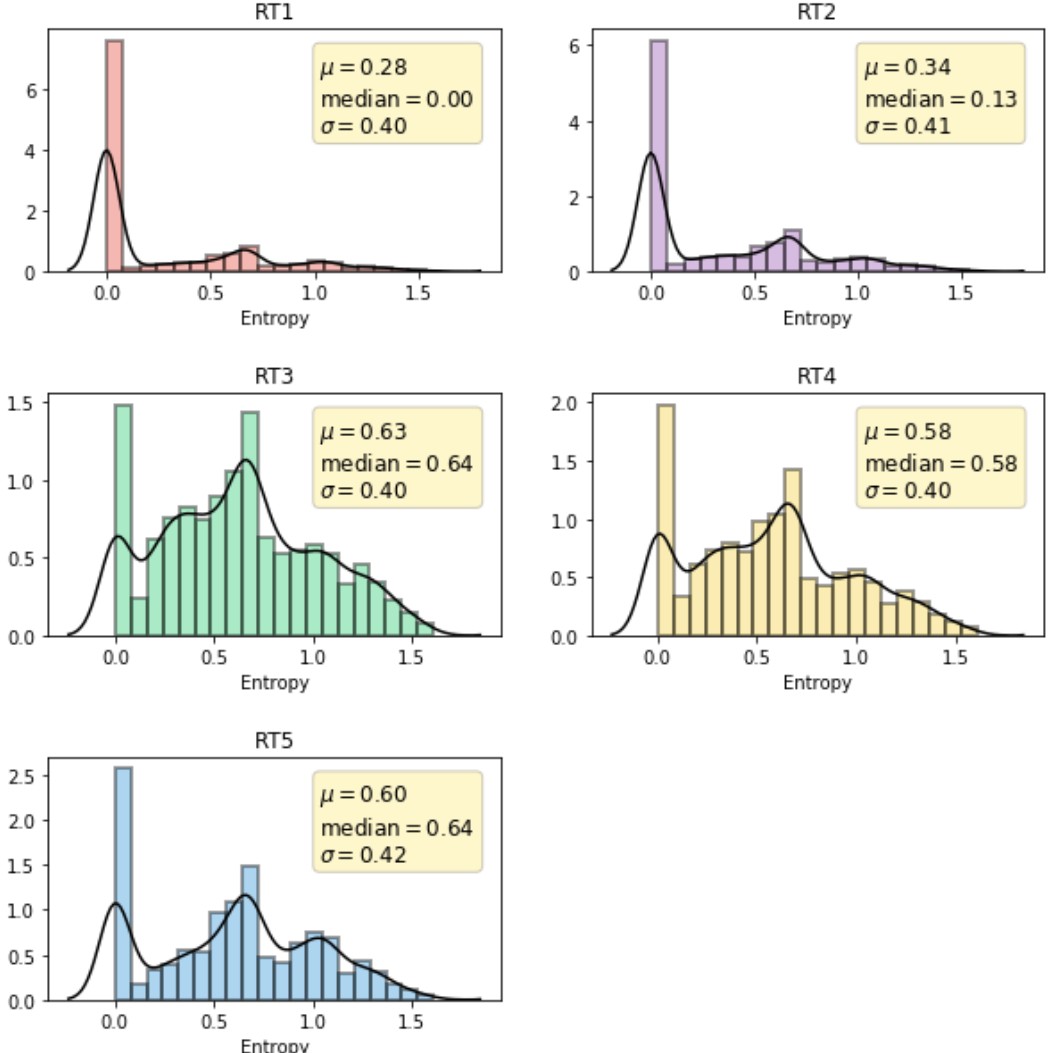

**Figure 11.** Distribution of the URL entropy for each community, fitted with a kernel density estimate.

## 4. Discussion

The Italian political discourse in the last few years has been dominated by the topic of migrations. In the previous sections, we analysed the national public debate on Twitter, with the goal of understanding how polarised the debate is and how the different factions can be characterised. We compared almost one year of Twitter data, filtered on specific keywords, to official information, gathered both from the Ministry of Interior and from the Italian Institute for Statistics. We geolocated more than 1.5 M tweets, and we compared the normalised geographical distribution of tweets with the normalised geographical distribution of long and short term residential permits to foreigners. The two distributions show a very low positive correlation, suggesting that the debate on Twitter might be disconnected from local realities while more related to the national political debate, supporting the hypothesis that people involved in a polarised debate are not particularly affected by external observations while they keep being involved in pointless words-fights [7,20].

The analysis of the network of Twitter interactions gave us very interesting insights on the structure of the debate. We focused on the networks of retweets, identifying communities of nodes sharing the same stance towards migration. These communities fit almost perfectly to the narrative of the Italian political landscape. Quite surprisingly, the influencers and political leaders that apparently lead the debate, do not necessarily belong to the clusters that include the majority of nodes. We find evidence of the existence of a 'silent majority' that is more connected to accounts who expose a more

positive stance toward migrants, while leaders whose stance is negative attract apparently more attention. We characterised these clusters not only by relying only on their central nodes with highest in-degree (i.e., by checking their most retweeted users), but we also labelled each user by its community in the retweet network and, for each user, we analysed all the bi-grams in the texts. This way we are able to understand whether the most used bi-grams suggest that the stance of the whole community is aligned to that of their central nodes. The outcome confirms us our hypothesis: the clusters' stances towards immigration look coherent with that of the central nodes, and we can clearly identify the pro-immigration and the anti-immigration groups, all centred around their political leaders and relevant news media. However, this debate is not represented by a clear 'pro' vs. 'against' dichotomy as with other divisive discussions that have been analysed (Democrats vs. Republicans [8–10], pro- vs. anti-vaxers [12,22], in favour vs. against referendums [11,20]. Instead, we found different clusters, apparently confirming a fragmented political scenario, even if two clearly opposite viewpoints emerge in the two extremes of the political spectrum.

We found a set of clusters showing heterogeneity in size, density and viewpoints. Interestingly, RT1 and RT2, the two communities linked to left-wing and right-wing areas, whose stance towards immigration is radically different, share a common feature: they are very closed on themselves, sharing low-entropy URLs, i.e., URLs that are scarcely shared by other communities, generating a sort of echo-chamber effect that keeps the content confined within groups of supposedly like-minded users.

All in all, our results suggest that the debate is driven and fuelled by politics and news media outlets and that it is strongly polarised and segregated, especially in light of the fact that communities display a low level of interaction. If the goal is to draw conclusions on someone's real stance towards immigration, Twitter might not be ideal: the discussion is strongly centred around politics, and political discourse on Twitter is continuously happening. This is true regardless of the specific topic: any important social and political news is discussed on Twitter by politicians, news media and common citizens. This means that users that are active in the debate about migration might be also active in other political debates, and they are likely to engage with the same people—their neighbours—no matter the topic under discussion. This introduces a form of noise: we might be looking more at the structure of the general political debate, rather than at the discussion about migration, also considered the relevance of the topic in the electoral campaigns and in the news. Nonetheless, the analysis of URLs diffusion gives us important information on the structure of the network. It confirms, as argued in [36], the importance of community structure—combined with social reinforcement and homophily—in the virality of content, because it might enable a sort of trapping effect that in this case we observed on the URLs.

## 5. Conclusions and Future Work

Our work shows that the Italian Twitter debate about migration is strongly segregated. The community structure affects the diffusion of content; regardless of the stance towards immigration, the clusters that can be linked to opposition parties are affected by a strong echo chamber effect, with the diffusing contents finding barriers at the boundaries of the clusters. There are a few points left open for discussion:

- We focused on the interactions between the five biggest communities, covering 87% of the nodes and leaving 29,969 out of the analysis. These remaining nodes gather in very small groups to form hundreds of micro-communities; therefore, it is harder for us to infer their stance towards migration as we did with the bigger clusters. Still, this does not necessarily mean that they do not play an important role in the debate. These nodes could be seen as the undecided—in our network they struggle to take a stance—and literature tells us that their role is often non trivial. Indeed, in [22], the authors find that anti-vaccination clusters are smaller numerically but are more central than others in terms of their positioning within the network, similarly to the behaviour of our clusters RT3 and RT4. They also find that the undecided are far from being passive. They are instead highly active and, with the system evolving, they get more and more entangled with the

anti-vaxers, i.e., most central groups that, while being a minority, manage to make a better use of the social media tool. Therefore our 29,969 undecided nodes, which are not taken into account in the present work, are worth being further explored, especially within the context of an evolving, rather than static, network of interactions.

- Both the community-wise hashtag and uni-grams analyses do not yield significant differences. This is partly because the topic is already filtered, specific and well defined, so there is not a large variety of hashtags; furthermore, the hashtag are strongly influenced by the external events, which we usually refer to with the same hashtag regardless of our stance towards them. Therefore, in this specific case, hashtags alone can hardly be used for topic or stance detection.

- Finally, we must remember that, even though we have more than 6 M tweets, we are still studying a restricted sample if compared to the true national debate. There is a number of biases that must be taken into account when dealing with social media data, first and foremost the fact that the population of users is most likely biased towards a certain socio-demographic cohort, as well as the fact that the behaviour of people on social media does not necessarily reflect their behaviour and ideas in "real life", especially when it comes to such delicate matters.

The analysis of the URLs and their association to an entropy measure leaves the door open for a detailed study of the diffusion of content in such a context, which is heavily segregated and biased from external influences such as political beliefs. The existence of local and global trends suggests that there might be room for a more comprehensive analysis, that focuses not only on the identification of echo chambers but also on the interplay between the network structure and the dynamics of complex contagion.

**Author Contributions:** Conceptualisation, D.P., G.R., S.V.; methodology, D.P., G.R., S.V., M.L.; formal analysis, S.V., M.L.; investigation, S.V., G.R., D.P., M.L.; data curation, S.V., D.P.; writing—original draft preparation, S.V.; writing—review and editing, S.V., G.R.; visualisation, S.V., M.L.; supervision, G.R., D.P. All authors have read and agreed to the published version of the manuscript.

**Funding:** Daniela Paolotti and Salvatore Vilella acknowledge support from the Lagrange Project of the Institute for Scientific Interchange Foundation (ISI Foundation) funded by Fondazione Cassa di Risparmio di Torino (Fondazione CRT).

**Conflicts of Interest:** The authors declare no conflict of interest.

## Abbreviations

| | |
|---|---|
| **FdI** | Fratelli d'Italia—Brothers of Italy |
| **FI** | Forza Italia—Go Italy |
| **Lega** | League—formerly known as Northern League |
| **LeU** | Liberi e Uguali—Free and Equal |
| **M5S** | Movimento 5 Stelle—5 Stars Movement |
| **PD** | Partito Democratico—Democratic Party |
| **RT** | Retweet |

## Appendix A. URLs Basic Statistics

Here, we report additional information about URLs that is not crucial to the paper but that could still be helpful to gain more context. In particular, we report in Table A1 the number of URLs with more than 100 shares in each community and, in Figure A1, the top 50 domains shared in the RT network.

**Table A1.** Number of URLs per community considered in the entropy calculation after data cleaning.

| ID | Number of URLs |
|----|----------------|
| RT1 | 16,659 |
| RT2 | 15,573 |
| RT3 | 4386 |
| RT4 | 5424 |
| RT5 | 6224 |

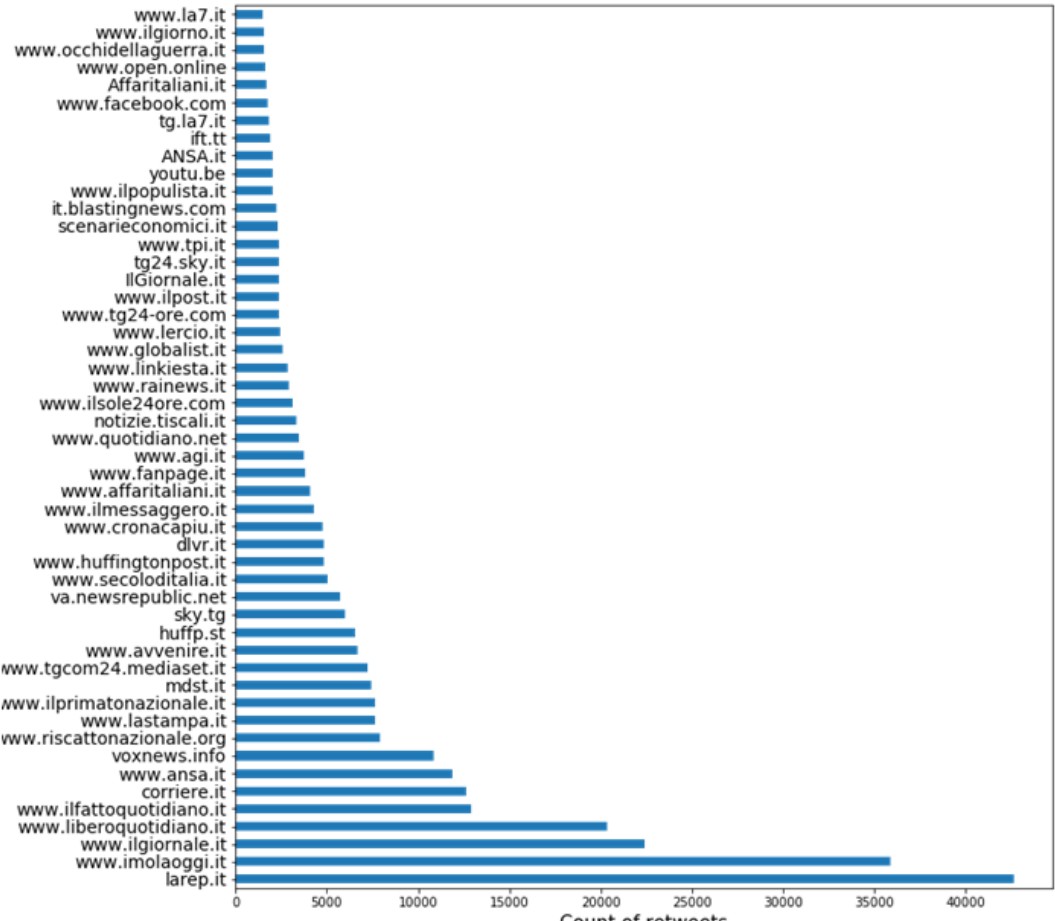

**Figure A1.** Top 50 domain names shared in the RT network.

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
