# Peer review of "Immigration as a Divisive Topic: Clusters and Content Diffusion in the Italian Twitter Debate"

_futureinternet, doi:10.3390/fi12100173_

Round 1

Reviewer 1 Report

The article deals with a really interesting topic: the debate on Twitter about immigration in Italy. Undoubtedly, the Italian case is especially relevant, because of the participation in the Government of a clearly right-wing political party, such as La Lega, with the Movimento 5 Stelle, which has never wanted to make it clear where it is situated on the ideological spectrum.

One of the strengths of this article is the number of messages that are analyzed. The authors state that they have analyzed more than 5 million tweets. Moreover, although there are other social media platforms with greater penetration among users, Twitter has been a good choice based on the objectives that the research raises.

However, although the work is correct, there are some points that should improve before being published. First, the statements made in the introduction are not supported by the previous literature, an aspect that should be resolved. There is a large number of investigations that deal with the position of the Italian parties regarding immigration, such as:

  • Chiaramonte, A., Emanuele, V., Maggini, N., and Paparo, A. (2018). Populist success in a Hung parliament: the 2018 general election in Italy. South European Society and Politics, 23, 479–501. doi: 10.1080/13608746.2018.1506513
  • Alonso-Muñoz, L., & Casero-Ripollés, A. (2020). Populism Against Europe in Social Media: The Eurosceptic Discourse on Twitter in Spain, Italy, France, and United Kingdom During the Campaign of the 2019 European Parliament Election. Frontiers in Communication, 5, 54. doi: 10.3389/fcomm.2020.00054
  • Caiani, M. (2019). The populist parties and their electoral success: different causes behind different populisms? The case of the Five-star Movement and the League. Contemporary Italian Politics, 11(3), 236-250. doi: 10.1080/23248823.2019.1647681

The importance of social media for the debate is also mentioned. In this sense, texts such as the following can be used by the authors:

  • Zheng, P., & Shahin, S. (2020). Live tweeting live debates: How Twitter reflects and refracts the US political climate in a campaign season. Information, Communication & Society, 23(3), 337-357.
  • Marcos-García, S., Alonso-Muñoz, L., & López-Meri, A. (2020). Extending influence on social media: The behaviour of political talk-show opinion leaders on Twitter. Communication & Society, 33(2), 277-293.
  • Mora-Cantallops, M., Sánchez-Alonso, S., & Visvizi, A. (2019). The influence of external political events on social networks: The case of the Brexit Twitter Network. Journal of Ambient Intelligence and Humanized Computing, 1-13.

Second, authors should bet on proposing 3 or 4 working hypotheses. Now key points are highlighted, but they are somewhat confusing. Posing hypotheses based on what has already been studied would make the investigation more rigorous.

Third, authors should review the structure of the text. Items 2.2 onwards should be part of a section 3. Results. Now it seems that the results of the research are part of the methodology used. If this change is done, current point 3 would become 4. Discussion and conclusions.

Four, regarding the Discussion and conclusions section, authors should make a greater effort to discuss their findings with those found by other researchers.

Also, Table 1 should be moved and presented in 2.1 and Figure 11 should be included in current 2.4 and not in the results section.

Reviewer 2 Report

I have carefully reviewed the manuscript, the topic may be of interest to the community of this Journal, however, I consider it important that at least the following aspects are revised

Q1. Figures 2-3, it is suggested to improve their quality, since the text is not very legible, that is, increase the font size of the axis and title labels.

Q2. Figure 8 and Figure A1 Increase 10% to improve text readability.

Q3. I suggest that the authors separate the analysis of results (discussion) from the conclusions. In other words, the Conclusions are another section of the paper.

Q4. I suggest that at the end of the Introduction, the authors write a paragraph indicating how the paper is organized.

Finally, after having meticulously reviewed the manuscript, I consider that the article only requires minor corrections.

Round 2

Reviewer 1 Report

The authors applied the changes proposed by the reviewers and improve the text notably. However it is necessary to check the list of references because the are some that are not well cited, for example, number 3.